# An Innovative Inducer of Platelet Production, Isochlorogenic Acid A, Is Uncovered through the Application of Deep Neural Networks

**DOI:** 10.3390/biom14030267

**Published:** 2024-02-23

**Authors:** Taian Yi, Jiesi Luo, Ruixue Liao, Long Wang, Anguo Wu, Yueyue Li, Ling Zhou, Chengyang Ni, Kai Wang, Xiaoqin Tang, Wenjun Zou, Jianming Wu

**Affiliations:** 1State Key Laboratory of Southwestern Chinese Medicine Resources, School of Pharmacy, Chengdu University of Traditional Chinese Medicine, Chengdu 611137, China; yitaian@stu.cdutcm.edu.cn (T.Y.); liyueyue@stu.cdutcm.edu.cn (Y.L.); 2Department of Chemistry, School of Basic Medical Sciences, Southwest Medical University, Luzhou 646000, China; ljs@swmu.edu.cn; 3Department of Pharmacology, School of Pharmacy, Southwest Medical University, Luzhou 646000, China; liaoruixue@stu.swmu.edu.cn (R.L.); wanglong1226@swmu.edu.cn (L.W.); wuanguo@swmu.edu.cn (A.W.); zhouling@stu.swmu.edu.cn (L.Z.); nichengyang@stu.swmu.edu.cn (C.N.); 20220599120055@stu.swmu.edu.cn (K.W.); 20210599120093@stu.swmu.edu.cn (X.T.); 4The Institute of Cardiovascular Research, Key Laboratory of Medical Electrophysiology of Ministry of Education, Luzhou 646000, China

**Keywords:** thrombocytopenia, deep learning, isochlorogenic acid A, hybrid neural networks, MK differentiation

## Abstract

(1) Background: Radiation-induced thrombocytopenia (RIT) often occurs in cancer patients undergoing radiation therapy, which can result in morbidity and even death. However, a notable deficiency exists in the availability of specific drugs designed for the treatment of RIT. (2) Methods: In our pursuit of new drugs for RIT treatment, we employed three deep learning (DL) algorithms: convolutional neural network (CNN), deep neural network (DNN), and a hybrid neural network that combines the computational characteristics of the two. These algorithms construct computational models that can screen compounds for drug activity by utilizing the distinct physicochemical properties of the molecules. The best model underwent testing using a set of 10 drugs endorsed by the US Food and Drug Administration (FDA) specifically for the treatment of thrombocytopenia. (3) Results: The Hybrid CNN+DNN (HCD) model demonstrated the most effective predictive performance on the test dataset, achieving an accuracy of 98.3% and a precision of 97.0%. Both metrics surpassed the performance of the other models, and the model predicted that seven FDA drugs would exhibit activity. Isochlorogenic acid A, identified through screening the Chinese Pharmacopoeia Natural Product Library, was subsequently subjected to experimental verification. The results indicated a substantial enhancement in the differentiation and maturation of megakaryocytes (MKs), along with a notable increase in platelet production. (4) Conclusions: This underscores the potential therapeutic efficacy of isochlorogenic acid A in addressing RIT.

## 1. Introduction

Cancer ranks as the second leading cause of death globally [1]. Despite the continuous advancements in medicine, there exist a multitude of methods for treating cancer [2]. Radiotherapy, a fundamental and highly efficacious approach, is employed in approximately 50% of cancer cases [3]. However, a significant challenge associated with radiotherapy is the development of thrombocytopenia as its most prevalent side effect. During radiotherapy, the proliferation of bone marrow (BM) hematopoietic cells is impeded, resulting in a rapid decline in peripheral blood (PB) platelet counts. This decline elevates the risk of infections and bleeding, potentially leading to treatment delays and unfavorable prognoses for cancer patients [4,5]. Current clinical strategies for managing thrombocytopenia involve platelet transfusions and drug therapy [6]. Nevertheless, prolonged infusion of platelets may induce various complications, including fever, splenomegaly, and allogeneic immunity [7]. Recombinant human thrombopoietin (rhTPO) and thrombopoietin receptor agonists (TPO-RA) are frequently utilized in clinical practice to stimulate the differentiation and maturation of MKs and enhance platelet production [8]. However, their use also carries an increased risk of venous thrombosis, which can have severe consequences for patients [9]. Consequently, there is a pressing need to find more effective, specific, and secure drugs to treat RIT.

Within the entirety of the chemical space, the pool of discoverable drug molecules is estimated to be approximately 10^60^ [10]. Unfortunately, traditional methods employed in drug discovery have struggled to keep pace with the escalating demand for novel pharmaceuticals. Statistical data reveal a consistent decline in the number of drugs approved by the US FDA over the years [11]. The conventional approaches to drug discovery involve protracted and expensive processes, characterized by multiple layers of screening. These methods necessitate the evaluation of millions to tens of millions of potential compounds, requiring numerous experiments for validation [12,13]. Recently, the utilization of artificial intelligence (AI) in drug research and development has seen substantial growth, propelled by advances in computing power and the availability of vast experimental datasets. AI applications span various stages of the research and development pipeline, encompassing target discovery, identification of seed compounds and lead compounds, and subsequent drug development and clinical trials. The integration of AI markedly enhances the efficiency of new drug discovery, simultaneously mitigating risks and reducing costs [12,13]. In the pursuit of multitarget inhibitors for idiopathic pulmonary fibrosis, Wang et al. devised an algorithm that seamlessly integrates network pharmacology, virtual screening, and machine learning. The efficacy of the identified compounds was subsequently assessed through rigorous in vitro and in vivo models [14]. Yang et al. proposed an innovative algorithm based on a flexible neural tree (FNT). This algorithm leverages the classification outcomes from multiple machine learning models as the input vector for FNT, employing a nonlinear integrated approach to pinpoint drug compounds associated with hypertension [15]. In a different domain, Sun et al. employed a CNN trained on peptide sequence coding to predict novel anticancer peptides characterized by selectivity and toxicity to cancer cells [16]. However, the exploration of artificial intelligence (AI) in the discovery of new drugs for the treatment of RIT has been relatively limited. In our previous work, our team utilized a naive Bayesian algorithm to construct a drug activity prediction model aimed at screening for compounds that can enhance MK differentiation and platelet production. This model identified methylophiopogonanone A as a potential therapeutic agent for RIT, demonstrating its ability to enhance MK differentiation and platelet production [17]. Additionally, our team screened wedelolactone, a compound promoting MK differentiation, using hybrid recurrent neural network (RNN) and DNN models [18].

In this study, we developed and employed models to predict potential therapeutic drugs for RIT with enhanced predictive performance. DL represents a rapidly advancing field in machine learning research [19]. Utilizing autoBioSeqpy (2.0 version) software [20], we initially employed two DL algorithms, namely CNN and DNN, along with two molecular fingerprints (MACCS and Morgan fingerprint), to construct an innovative prediction model. To further improve the predictive accuracy, we integrated the learning features of the CNN and DNN models, creating a hybrid neural network model. The reliability of this model was assessed through testing with 10 FDA-approved drugs for thrombocytopenia. Subsequently, the mixed DL model was deployed to predict the activity of 2070 compounds within the Chinese Pharmacopoeia Natural Product Library. Potential candidates with the activity to promote differentiation and maturation of MKs and production of platelets were identified, and 39 compounds were subjected to further verification. Among these, isochlorogenic acid A demonstrated a notable capacity to promote MK differentiation and maturation, displaying high induction activity for MK differentiation and maturation in K562 and Meg-01 cells. Then, in vivo experiments further substantiated the significant therapeutic effect of isochlorogenic acid A on RIT mice.

## 2. Materials and Methods

### 2.1. Dataset Collection and Processing

Our training dataset comprised a total of 781 compounds sourced from study reports and laboratory screenings. Based on their impact on MK differentiation and platelet production, 38 compounds were categorized as inducers, and the remaining 743 compounds were labeled as noninducers. Since the number of inducers was significantly lower than that of noninducers, the imbalance in sample ratios could result in overfitting of the model during training. To address this issue, we employed SMILES enumeration [21] to expand the positive samples to rectify the imbalanced dataset. To calculate compound similarities, the Tanimoto coefficient [22] was employed. Subsequently, the MaxMin algorithm in rdkit was utilized to eliminate data with high similarity, thereby enhancing the diversity of the dataset [23].

### 2.2. Construction, Training, and Evaluation of Models

In this study, we trained three neural network-based models, namely a DNN model, CNN model, and HCD model, to predict compound activity, leveraging molecular fingerprints as descriptor features. We employed two widely recognized fingerprinting methods: the CNN model received Molecular Access System (MACCS) key fingerprints, comprising 166 public keys [24]. In contrast, the DNN model utilized Morgan fingerprints, also known as circular fingerprints—these are circular substructure fingerprints with a specified radius of 2 and a size of 2048 bits [25]. The molecular fingerprint for both inputs was computed using DeepChem (2.6.0.dev20210622012521 version) software (https://deepchem.io/ (accessed on 21 July 2023)). In addition, we trained five machine learning models using orange3 (3.26.0 version) (https://orangedatamining.com/download/ (accessed on 29 July 2023)): Support Vector Machine (SVM), Random Forest (RF), Neural Network (NN), Logistic Regression (LR), and Adaptive Boosting (AB). We used MACCS keys fingerprint, Morgan fingerprint, and a combination of MACCS key fingerprints with Morgan fingerprints as their inputs.

A CNN is a type of neural network that employs convolutional operations instead of fully connected layers. In the CNN model, the first layer is the embedding layer, which is responsible for converting positive integers (index values) into dense vectors of a fixed size. Subsequently, the convolutional layer is the foundational element for constructing convolutional neural networks. This layer generates a convolution kernel applied to the layer’s inputs in a single spatial (or temporal) dimension, resulting in an output tensor. The parameters for the convolutional layer include a convolution window length of 16, a stride of 1, and the utilization of 330 filters. Following the convolutional layer, there is a pooling layer for maximum pooling of data, succeeded by a dense layer featuring 128 neurons and a rectified linear unit (ReLU) activation function. The embedded layer and the dense layer use a dropout rate of 20% to mitigate overfitting. The output layer consists of a single neuron activated by a sigmoid function. On the other hand, the DNN model is composed of three dense layers interleaved with dropout layers, utilizing the ReLU activation function and a 20% dropout rate. The number of neurons in each dense layer is 2048, 64, and 16, respectively. The final output layer is activated by a sigmoid function. For hybrid models, post-training, the output layers of both the CNN and DNN models are removed. The output of the remaining layers is connected and fed into the unified dense layer, which is activated by the sigmoid function. Detailed workflow diagrams for the CNN, DNN, and HCD models are shown in Appendix A.

For the development and implementation of the model, we employed the DL tool autoBioSeqpy (2.0 version) (https://github.com/jingry/autoBioSeqpy (accessed on 9 August 2023)). All models underwent 20 epochs of training iterations with a batch size of 32 on a personal computer. The training, utilizing the Adam optimizer, aimed to minimize the binary cross-entropy loss between the target and predicted output. The learning rate for the optimizer was consistently set at 0.001 for all epochs. Each model underwent the training and testing process three times, and the output is the average result of those iterations. Throughout the training and testing phases, 80% of the data were randomly allocated for training and validation, with the remaining 20% reserved for testing.

The performance evaluation of the developed models was conducted using several metrics, including accuracy (*ACC*), precision (*PRE*), *recall*, *F-value*, and Matthews correlation coefficient (*MCC*). Additionally, receiver operating characteristic (ROC) and precision recall curves (*PRC*) were plotted, and the corresponding values of the area under the curve, namely AUROC and AUPRC, were calculated to compare the screening performance of these models. The following equations were employed to calculate *ACC*, *PRE*, recall, *F-value*, and *MCC*:
$$ACC=\frac{TP+TN}{TP+FP+TN+FN}$$


$$PRE=\frac{TP}{TP+FP}$$


$$Recall=\frac{TP}{TP+FN}$$


$$F\text{-}value=2\times \frac{TP}{2TP+FP+FN}$$


$$MCC=\frac{\left(TP\times TN)-(FN\times FP\right)}{\sqrt{\left(TP+FN)\times (TN+FP)\times (TP+FP)\times (TN+FN\right)}}$$

where *TP*, *FP*, *FN*, and *TN* are true positive, false positive, false negative and true negative, respectively.

### 2.3. Chemicals

The Chinese Pharmacopoeia Natural Product Library was obtained from TargetMol Chemicals Inc. (Boston, MA, USA). Isochlorogenic acid A (purity = 98.83%, as determined by HPLC) was purchased from Chengdu Pufei De Biotech Co., Ltd. (Chengdu, China) and reconstituted following the product instructions.

### 2.4. Cell Culture

The human chronic myeloid leukemia (CML) cell line K562 and the human megakaryoblastic leukemia cell line Meg-01 were purchased from American Type Culture Collection (Rockville, MA, USA). The two cell lines were cultured in RPMI 1640 medium (Gibco Life Technologies, Carlsbad, CA, USA) supplemented with 10% fetal bovine serum (FBS, CAT: SP10020500, Sperikon Life Science & Biotechnology Co., Ltd., Chengdu, China) and 1% penicillin–streptomycin solution (Gibco, Invitrogen Corporation, Carlsbad, CA, USA) in a humidified atmosphere with 5% CO_2_ at 37 °C.

### 2.5. LDH Assay

K562 and Meg-01 cells were plated in 96-well plates (5 × 10^3^ per well). After treatment with or without ICGA-A (5, 10, and 20 μM), the cells were cultured for 2 days, 4 days, and 6 days. The LDH assay was performed using an LDH assay kit (Beyotime, Haimen, China) following the manufacturer’s instructions. 

### 2.6. Morphological Analysis 

Cells were seeded in 12-well plates at a density of 2 × 10^4^ cells/mL. After treatment with or without ICGA-A (5, 10, and 20 μM) for 5 days, the cell morphology was observed using a microscope (NIKON, Tokyo, Japan) at 10× resolution.

### 2.7. Giemsa Staining

K562 and Meg-01 cells were plated in 12-well plates (2 × 10^4^ per well) and treated with or without ICGA-A (5, 10, and 20 μM). On the fifth day, the cells were washed and harvested with phosphate buffered saline (PBS) and swelled with 0.075 M KCl solution. Then, the cells were fixed with fixing solution (methanol: glacial acetic acid = 3:1 (*v*/*v*)), placed onto glass slides and stained with Giemsa solution (Solarbio, Beijing, China) for 10 min at room temperature. The cells were washed with distilled water and viewed under a microscope (NIKON, Tokyo, Japan).

### 2.8. Phalloidin Staining 

K562 and Meg-01 cells were plated in 12-well plates (2 × 10^4^ per well) and treated with or without ICGA-A (5, 10, and 20 μM). On the fifth day, cells were resuspended in PBS through a TD3 cytocentrifuge (Shanghai Lu Xiangyi Centrifuge Instrument Co., Ltd., Shanghai, China) and harvested onto glass slides. The cells were then fixed with a 4% paraformaldehyde solution (Biosharp, Hefei, China) for 12 min and then permeabilized with 0.5% Triton X-100 for 4 min. After being washed three times with PBS, the cells were incubated with a working solution of TRITC-labeled phalloidin (1:200) (Solarbio, Beijing, China) for 30 min in the dark at room temperature. Then, DAPI (Solarbio, Beijing, China) was added to counterstain the nucleus for 1 min. The images were captured using an inverted fluorescence microscope (Nikon Ts2R/FL, Japan).

### 2.9. Analysis of MK Differentiation 

K562 and Meg-01 cells treated with or without ICGA-A (5, 10, and 20 µM) for 5 days were harvested and washed with ice-cold PBS 2 times. Then, 100 µL of the cell suspension at a density of 1.0 × 10^6^ cells/mL was transferred to l mL Eppendorf tubes, followed by incubation with FITC-conjugated anti-CD41 and PE-conjugated anti-CD42b antibodies (Biolegend, San Diego, CA, USA) for 30 min on ice in the dark. The expression of CD41 and CD42b was detected using a BD FACSCanto II flow cytometer (BD Biosciences, San Jose, CA, USA), and the results were analyzed using FlowJo software (10.9 version).

### 2.10. Animals 

Specific pathogen-free (SPF) Kunming (KM) mice, aged 8–10 weeks and weighing 18–22 g, were purchased from Da-shuo Biotechnology Limited (Chengdu, Sichuan, China). The mice were housed in a controlled facility with an ambient temperature of 25–26 °C and a relative humidity range of 50% to 60%. They were kept under a 12 h/12 h light/dark cycle and were fed a standard mouse diet along with purified water. 

### 2.11. Construction of the RIT Model and ICGA-A Treatment

After 1 week of feeding the KM mice, all mice were randomly assigned into 6 groups. The groups included a normal control group, a RIT model group, a rhTPO (3SBIO; Shenyang, China, 3000 U/kg, positive control) group, a low-dose ICGA-A (5 mg/kg) group, a medium-dose ICGA-A (10 mg/kg) group, and a high-dose ICGA-A (20 mg/kg) group with 6 males and 6 females in each group. In addition to the normal control group, all mice in the other groups were given a single dose of 4 Gy X-rays for total body irradiation to establish a mouse model of RIT. After irradiation, the mice in the normal control and RIT model group were intraperitoneally injected with normal saline per day. The mice in the positive control group were intraperitoneally injected with rhTPO (3000 U/kg) daily. All mice in the ICGA-A treatment group were intraperitoneally injected with 5 mg/kg, 10 mg/kg, and 20 mg/kg of ICGA-A daily for 10 consecutive days, respectively.

### 2.12. Measurement of Hematologic Parameters 

A small amount of PB (40 μL) was drawn from the fundus vein plexus of all mice and treated with 160 μL of diluent on days 0, 3, 7, and 10. Hematological parameters were then measured using an automatic hematology analyzer (Sysmex XT-2000iV, Kobe, Japan).

### 2.13. Measurement of Body Weight and Visceral Index 

The mice were weighed on day 0, 3, 7, and 10 of administration and euthanized by cervical dislocation on day 10. The liver and kidneys were extracted and weighed, and organ indices were calculated.

### 2.14. Histology Analysis 

After mice were treated with ICGA-A for 10 days, three mice were randomly selected from each group, and the femur kidneys, spleens, and livers were completely infiltrated in 10% paraformaldehyde for 24 h. The femurs were then decalcified with decalcification solution for more than 1 month. The organs were then embedded in paraffin, cut into 5 µm sections, stained with hematoxylin and eosin (H&E), and photographed under an Olympus BX51 microscope (Olympus Optical, Tokyo, Japan). The field of view for each sample was randomly selected, and the number of MKs was counted.

### 2.15. Flow Cytometry Analysis of PB Cells 

To analyze platelets in PB, 50 µL of blood was randomly collected from the eye venous cluster of three mice and mixed with 1 mL of sodium citrate buffer. The samples were then labeled with 0.5 µL of FITC-conjugated anti-CD41 (BioLegend, San Diego, CA, USA), 1.25 µL of PE-conjugated anti-CD61 (BD Biosciences, San Jose, CA, USA), 0.5 µL of FITC-conjugated anti-CD41, and 1.25 µL of PE-conjugated anti-CD62P (BioLegend, San Diego, CA, USA) on ice for 15 min in the dark. After resuspension in 300 µL PBS, flow cytometry was performed using a BD FACSCanto II flow cytometer (BD Biosciences, San Jose, CA, USA), and the results were analyzed using FlowJo software.

### 2.16. Flow Cytometry Analysis of BM and Spleen Cells

To analyze the MKs in the BM and spleen, the total BM cells from the femur were rinsed with normal saline. The spleen was ground into single cells and filtered using a nylon mesh. The red blood cells (RBCs) from the cell sample were then removed using a red blood cell lysis buffer (Beijing 4 A Biotech, Beijing, China). These cells were collected in 100 µL of PBS and labeled with 0.5 µL of FITC-conjugated anti-CD41 and 1.25 µL of PE-conjugated anti-CD61 on ice for 15 min in the dark. Finally, after resuspension in 300 µL of PBS, flow cytometry was performed using a BD FACSCanto II flow cytometer (BD Biosciences, San Jose, CA, USA), and the results were analyzed using FlowJo software.

### 2.17. Polyploidy Analysis of BM and Spleen Cells

Cells from the BM and spleens were labeled with 0.5 µL of FITC-conjugated anti-CD41 monoclonal antibody on ice for 15 min in the dark. They were then incubated with PI stain solution for 15 min on ice in the dark. Polyploidy was analyzed using a BD FACSCanto II flow cytometer (BD Biosciences, San Jose, CA, USA), and the results were analyzed using FlowJo software.

### 2.18. Statistical Analysis

All of the results in this study are presented as the mean standard deviations (SDs) of at least three independent experiments. Statistical significance among groups was assessed using analysis of variance (ANOVA) in GraphPad Prism 9.0 software to determine statistical significance. A *p*-value of 0.05 was used as the threshold for statistical significance.

## 3. Results

### 3.1. Data Collection and Processing

SMILES enumeration is the process of writing out all possible forms of a molecule using SMILES symbols. It is a useful technique for enhancing data prior to sequence-based molecular modeling [26]. We amplified SMILES 20-fold to 760 for 38 positive compounds. The pairwise Tanimoto coefficients between positive and negative samples were then calculated separately (Figure 1A,B). Then, the MaxMin algorithm in RDKit was used to extract the 500 samples with the lowest pairwise Tanimoto coefficients between the positive and negative datasets, resulting in two balanced and diverse datasets (Figure 1C,D). After removing the redundant data, we could observe a significant reduction in the red parts with high similarity. The mean Tanimoto coefficients decreased from 0.275 to 0.152 for positive samples and from 0.217 to 0.125 for negative samples.

### 3.2. Development of Deep Learning Models for Drug Screening for the Treatment of RIT Using autoBioSeqpy

AutoBioSeqpy is a tool designed for the classification of biological sequences using deep learning techniques. Its distinct advantage lies in its simplicity, making it particularly suitable for users who may not have extensive expertise in deep learning. Users only need to prepare the input dataset and then utilize the computer’s command interface to input the appropriate commands. AutoBioSeqpy automates a series of customizable steps, including text reading, parameter initialization, sequence encoding, and model loading, training, and evaluation. The tool has been upgraded to version 2.0, expanding its support for various data types and hybrid model structures. Here is a brief demonstration of how to build, train, test, evaluate, and apply DL models in autoBioSeqpy for drug activity prediction tasks, utilizing the following command-line commands for the CNN, DNN, and HCD models.

CNN model:
python running.py --dataType other --dataEncodingType other --dataTrainFilePaths examples/ICGA-A/data/pos_train.txt examples/ICGA-A/data/neg_train.txt --dataTrainLabel 1 0 --dataSplitScale 0.8 --modelLoadFile examples/ICGA-A/model/CNN.py --verbose 1 --outSaveFolderPath tmpOut --savePrediction 1 --saveFig 1 --batch_size 32 --epochs 20 --shuffleDataTrain 1 --spcLen 167 --modelSaveName tmpMod.json --weightSaveName tmpWeight.bin --noGPU 1 --paraSaveName parameters.txt --optimizer optimizers.Adam(lr=0.001,amsgrad=False,decay=False)

To initiate autoBioSeqpy using Python (3.8.8 version) (python running.py), specify the input molecule compounds as feature vectors (--dataType other). By utilizing the code (--dataEncodingType other), autoBioSeqpy will disable the default encoding method and directly read externally calculated features. Input the training set into the software, comprising both positive and negative training sets (--dataTrainFilePaths examples/ICGA-A/data/pos_train.txt examples/ICGA-A/data/neg_train.txt), with positive samples labeled as 1 and negative samples labeled as 0 (--dataTrainLabel 1 0). With the code (--dataSplitScale 0.8), autoBioSeqpy randomly divides the input samples into an 80% training-validation set and a 20% test set. Load the CNN model (--modelLoadFile examples/ICGA-A/model/CNN.py). Output log information with a progress bar (--verbose 1). The string length was set to 167 (--spcLen 167), consistent with the length of the MACCS key fingerprint. Additionally, the order of the training data (--shuffleDataTrain 1) is shuffled to prevent overfitting and ensure that the data are different for each model call. After execution, all result files are stored in the “tmpOut” folder (outSaveFolderPath tmpOut).

DNN model:
python running.py --dataType other --dataEncodingType other --dataTrainFilePaths examples/ICGA-A/data/pos_train_feature.txt examples/ICGA-A/data/neg_train_feature.txt --dataTrainLabel 1 0 --dataSplitScale 0.8 --modelLoadFile examples/ICGA-A/model/DNN.py --verbose 1 --outSaveFolderPath tmpOut --savePrediction 1 --saveFig 1 --batch_size 32 --epochs 20 --shuffleDataTrain 1 --spcLen 2048 --modelSaveName tmpMod.json --weightSaveName tmpWeight.bin --noGPU 1 --paraSaveName parameters.txt --optimizer optimizers.Adam(lr=0.001,amsgrad=False,decay=False)

The DNN model input is a circular fingerprint with a length of 2048 bits (--spcLen 2048).

HCD model:
python running.py --dataType other other --dataEncodingType other other --dataTrainFilePaths examples/ICGA-A/data/pos_train.txt examples/ICGA-A/data/neg_train.txt examples/ICGA-A/data/pos_train_feature.txt examples/ICGA-A/data/neg_train_feature.txt --dataTrainLabel 1 0 1 0 --dataSplitScale 0.8 --modelLoadFile examples/ICGA-A/model/CNN.py examples/ICGA-A/model/DNN_hybrid.py --verbose 1 --outSaveFolderPath tmpOut --savePrediction 1 --saveFig 1 --batch_size 64 --epochs 20 --shuffleDataTrain 1 --spcLen 167 2048 --modelSaveName tmpMod.json --weightSaveName tmpWeight.bin --noGPU 1 --paraSaveName parameters.txt --optimizer optimizers.Adam(lr=0.001,amsgrad=False,decay=False) --dataTrainModelInd 0 0 1 1

Input program (--dataTrainModelInd0 0 1 1); autoBioSeqpy can specify that the first two datasets are trained by model_0 (CNN model) and the last two datasets are trained by model_1 (DNN model).

Upon completion of model training, autoBioSeqpy generates five model evaluation indicators using the test set, including ACC, PRE, recall, F-value, and MCC. Additionally, the tool produces ROC and PR curve plots to assess the model, as illustrated in Table 1 and Figure 2. All models demonstrate excellent classification performance. The DNN model achieved AUROC and AUPRC values of 0.979 and 0.973, respectively, surpassing the CNN model’s values of 0.975 and 0.965. Notably, the HCD model exhibited superior performance, boasting the highest AUROC and AUPRC scores among all models at 0.994 and 0.992. Moreover, the remaining five metrics—ACC (98.3%), PRE (97.0%), recall (99.8%), F-value (98.3%), and MCC (0.967)—were also the highest across the three models. In addition, the ACC, PRE, recall, F-value, and AUROC of the deep learning model HCD were all higher than those of the five machine learning models (Appendix A). This outcome suggests that the CNN and DNN models learn distinct features from the dataset, and their combination in the HCD model enhances the ability to differentiate drug activity. 

### 3.3. Verification of the Feasibility of the HCD Model for Virtual Screening to Promote Platelet Production Drugs

We collected 10 FDA-approved drugs for the treatment of human thrombocytopenia. After developing, training, and evaluating the model, we applied the best HCD model to virtually validate the activity of the compounds using the following command line:
python predicting.py --paraFile tmpOut/parameters.txt --dataTestFilePaths examples/ICGA-A/data/FDA.txt examples/ICGA-A/data/FDA_feature.txt --predictionSavePath tmpout/indPredictions.txt --dataTestModelInd 0 1

In this model, each molecule is assigned a prediction score ranging from 0 to 1, and the decision threshold is set at 0.5. Compounds with a score above 0.5 are classified as drugs that promote platelet production. According to this criterion, the model predicted seven out of 10 drugs known to promote platelet production as active (Figure 3), while the machine learning model could predict that up to two drugs would be active (Appendix A). These results highlight the outstanding capability of the HCD model in identifying compounds with the potential to promote platelet production. 

### 3.4. Virtual Screening for Potentially Promoted Platelet Production Drugs Using the HCD Model 

We utilized the following command line to identify potential platelet-stimulating drugs from the Chinese Pharmacopoeia Natural Product Library:
python predicting.py --paraFile tmpOut/parameters.txt --dataTestFilePaths examples/ICGA-A/data/CPNP-Library.txt examples/ICGA-A/data/CPNP-Library_feature.txt --predictionSavePath tmpout/indPredictions.txt --dataTestModelInd 0 1

We selected 2070 natural products from the Chinese Pharmacopoeia library that contain the main active ingredients of 246 traditional Chinese medicines such as Salvia miltiorrhiza, Eucommia, and Astragalus. Our model predicted that 39 of these compounds have activities that promote MK differentiation and platelet promotion. After treating cells with them for 5 days, the cells were observed with an inverted microscope. It was shown that out of the 39 predicted compounds, only three (ingenol, abscisic acid, and isochlorogenic acid A) exhibited high activity (Figure 4). Ingenol has been reported to promote platelet production and MK differentiation [27]. Abscisic acid has also been reported as an inducer of MK differentiation and platelet production [28].Yet, studies on the use of isochlorogenic acid A in the treatment of thrombocytopenia have not been reported. In addition, we calculated a Tanimoto Coefficient between the compound in the training set and ICGA-A (Appendix A). We did not find highly similar compounds in the training set. A UMAP analysis was then conducted, revealing that ICGA-A was similar to the training set in chemical space defined by the MACCS keys and Morgan fingerprint (Appendix A). This finding demonstrates that deep learning models can uncover the underlying structure of compounds. The above results show that ICGA-A is a novel compound with the potential to promote megakaryocyte differentiation and platelet formation.

### 3.5. Verification of the Activity of Compounds Promoting MK Differentiation

The HCD model predicted 39 compounds with potential activity from the Chinese Pharmacopoeia Natural Product Library. We selected K562 cells and Meg-01 cells for cellular-level activity validation. After treating K562 cells and Meg-01 cells with these compounds for 5 days (one of the characteristics of MKs is an enlarged nucleus [29]), we found that there were many large cells in the ICGA-A (5, 10, and 20 μM)-treated group, while the number of large cells in the control group without ICGA-A treatment was very low (Figure 5A). When K562 cells and Meg-01 cells were treated with other compounds, no large cells were observed under the microscope. These data suggest that only compound ICGA-A, which was virtually screened by the HCD model, may promote MK differentiation.

### 3.6. ICGA-A Induces MK Differentiation and Maturation of K562 and Meg-01 Cells

The differentiation and maturation of MKs are important stages in platelet production. This process is characterized by cell enlargement, nuclear polyploidization, and increased expression of markers specific to megakaryocytic cells [30]. To investigate the impact of ICGA-A on MK differentiation of K562 and Meg-01 cells, we treated them with or without different concentrations of ICGA-A (5, 10, and 20 μM) for 5 days. Afterward, we performed Giemsa staining, which revealed the presence of large cells with multiple nuclei in the ICGA-A-treated group. In contrast, the control group only showed small monocytes (Figure 5B). Phalloidin staining showed that the cells in the ICGA-A-treated group were enlarged, multilobed, and multinucleated, while the nuclei in the control group were less differentiated. In addition, treatment with ICGA-A significantly increased the expression and induced an aggregation of F-actin, which may be beneficial for proplatelet formation (Figure 5C,D). Then, we examined the expression of specific surface antigens CD41 and CD42b on MKs using flow cytometry. After ICGA-A treatment, the proportion of CD41^+^CD42b^+^ cells in K562 and Meg-01 cells significantly increased in a concentration-dependent manner compared to that in the control group (Figure 5E–H). These results indicated that ICGA-A promoted the differentiation of K562 and Meg-01 cells into MKs. In addition, the cytotoxicity of ICGA-A was assessed using the lactate dehydrogenase (LDH) assay. We found no significant difference in the amount of LDH released between K562 and Meg-01 cells in all ICGA-A treatment groups and the control group (Figure 5I,J). The above results suggest that ICGA-A can promote the differentiation and maturation of MKs without causing cytotoxicity.

### 3.7. ICGA-A Promoted Platelet Production and Increased the Number of MKs in RIT Mice

Because ICGA-A promotes MK differentiation and maturation of K562 and Meg-01 cells, we utilized the RIT mouse model to further investigate the in vivo efficacy of ICGA-A. Saline (10 mL/kg), rhTPO (3000 U/kg), and ICGA-A (5, 10, and 20 mg/kg) were administered intraperitoneally daily for a period of 10 days. Routine blood tests were performed on days 0, 3, 7, and 10, and we found that platelet numbers in irradiated mice were significantly higher than those in the model group on days 7 and 10 after ICGA-A and rhTPO treatment (Figure 6A), suggesting that ICGA-A promotes platelet production. The numbers of white blood cells (WBCs) and red blood cells (RBCs) were also measured. The level of leukocytes in the PB of mice decreased rapidly after irradiation. However, there was no significant difference observed between the model group, ICGA-A treatment group, and rhTPO treatment group (Figure 6B). This indicates that the RIT model was successfully established. There was no significant difference in the number of RBCs among the model group, ICGA-A treatment group, and rhTPO treatment group (Figure 6C). The results of routine blood tests indicated that ICGA-A played a significant role in platelet recovery without inhibiting the production of WBCs and RBCs. From day 7, the ICGA-A and rhTPO-treated groups showed a significant increase in body weight compared to the model group, indicating the protective effect of ICGA-A against radiation-induced body weight loss (Figure 6D). Visceral indices of the kidney and liver were also measured, and no significant differences were found between the groups of mice (Figure 6E,F). In addition, we performed H&E staining on the mouse livers and kidneys to examine whether ICGA-A could cause organ damage. The results showed that there were no significant pathological changes in the tissue sections of the ICGA-A treatment group compared with the control group (Figure 6G,H). Taken together, these results demonstrate that ICGA-A can promote platelet production without any side effects in mice.

Platelets are produced by MKs in hematopoietic tissue [31]. Therefore, we investigated the effect of ICGA-A on the generation of MKs. H&E staining was used to determine whether ICGA-A promoted the formation of MKs in tissues. The results showed that the number of MKs in the spleen and BM of the mice in the ICGA-A and rhTPO treatment groups was significantly higher than that in the model group (Figure 7A–D), indicating that ICGA-A could promote the production of MKs in the spleen and BM. The expression of CD41 and CD117 on BM cells was then detected by flow cytometry. The results showed that the levels of hematopoietic progenitor cells (CD41^−^CD117^+^), MK progenitor cells (CD41^+^CD117^+^), and MKs (CD41^+^CD117^−^) in the ICGA-A and rhTPO treatment groups were significantly higher than those in the model group (Figure 7E,F). This suggests that ICGA-A promotes the generation of hematopoietic progenitor cells and their differentiation into MK progenitor cells and MKs. Then, the surface antigens CD41 and CD61, known as MK differentiation markers, were detected on BM and spleen cells. The results showed that the proportion of CD41^+^CD61^+^ cells in BM and spleen cells in the ICGA-A and rhTPO treatment groups was significantly higher than that in the model group (Figure 7G–J). This suggests that ICGA-A promoted the differentiation of MKs in the BM and spleen. Through cytoploidy experiments, we found that the proportion of 2 N cells in the BM and spleen of the ICGA-A and rhTPO treatment groups was significantly lower than that in the model group, while the proportion of 4 N and ≥8 N cells was significantly higher than that in the model group (Figure 7K–N), and the results showed that ICGA-A promoted the maturation of MKs in the BM and spleen. The above results show that ICGA-A promotes the differentiation and maturation of MKs in the BM and spleen.

In addition, we performed flow cytometry analysis of PB cells using CD41, CD61, and CD62P markers. The results showed that the proportion of CD41^+^CD61^+^ cells in the ICGA-A and rhTPO treatment groups was significantly higher than that in the model group (Figure 8A,B), indicating that ICGA-A increased the PB platelet count. CD62P (platelet surface P-selectin) is a marker of activated platelets. The results showed that the percentage of activated platelets (CD41^+^CD62P^+^) in the ICGA-A and rhTPO treatment groups was significantly lower than in the model group and similar to that in the control group (Figure 8C,D). This suggests that ICGA-A treatment can improve abnormal platelet activation in HIT mice. In summary, ICGA-A promotes the differentiation and maturation of MKs, alleviates abnormal platelet activation, and has no side effects. It has the potential to be a therapeutic treatment for RIT.

## 4. Discussion

Thrombocytopenia, a common complication of cancer treatment, can result in bleeding, delayed treatment, and even death [4,32,33]. Patients with RIT can stop bleeding or prevent bleeding through platelet transfusions, but this treatment is associated with side effects such as fever, allergies, bacterial infections, and transfusion-related acute lung injury [34]. Currently, there is no specific drug for RIT. The FDA has approved the use of TPO-RAs for the treatment of immune thrombocytopenia (ITP). However, their clinical use is limited due to toxic side effects [35,36]. Therefore, there is an urgent need to discover more effective and safer drugs to treat of RIT. However, the development of new drugs is a challenging and expensive process that demands significant resources and time, and it also has a remarkably high rate of failure [37]. There are an estimated 10^60^ drug-like molecules in the theoretical expansion of chemical space beyond known compounds [38]; only a small number of compounds bind to specific parts of the body and, as a result, exert biological activity. Therefore, determining how a molecule acts on a specific target is a highly complex task [39]. The research and development of traditional drugs involves the collaboration of numerous researchers. They screen through millions, or even tens of millions, of potential compounds in a step-by-step process. This requires conducting a large number of experiments over an extended period of time to verify the drug activity of these compounds [40]. At present, the emergence of artificial intelligence (AI) offers new possibilities for addressing the challenges associated with high costs, long development cycles, high risks, and low success rates in traditional drug development [41]. Machine learning (ML) is a branch of artificial intelligence commonly used in the pharmaceutical industry for virtual screening of active compounds [42]. In this study, we first developed a drug virtual screening model using a deep learning (DL) algorithm, which is a subfield of machine learning that aims to train models by emulating human learning processes [41,43]. This virtual screening model was used to predict potential natural compounds that promote MK differentiation and platelet production. Subsequently, their drug activity was validated through a series of cellular and animal experiments.

The dataset used in this study consisted of 38 active compounds and 743 inactive compounds. Due to the significant disparity in the number of active compounds and inactive compounds, we amplified the positive samples during training to balance the positive and negative sample ratios in the dataset. This helped prevent overfitting of the model. Then, we calculated the similarity between the compounds, removed the data with high similarity, and enhanced the diversity of the dataset. The optimized data were fed into the DL model, and after evaluation and testing, the best model was selected to predict the activity of 2070 compounds in the Chinese Pharmacopoeia Natural Product Library.

We built three deep learning models: CNN, DNN, and Hybrid CNN+DNN (HCD). Convolutional neural networks (CNNs) have structural characteristics such as local area connections and weight sharing. Weight sharing makes the network structure of CNNs more like biological neural networks, while local connections differ from traditional neural networks. In a CNN, each neuron in the nth layer is connected to all neurons in layer n-1, but only to some neurons in the nth layer. The effect of these two characteristics is to reduce the complexity of the network model and decrease the number of weights [44,45]. A deep neural network (DNN) is a multilayered unsupervised neural network consisting of an input layer, one or more hidden layers, and an output layer. The layers are fully connected, meaning that the features from one layer are inputted to the next layer. This allows for feature learning, which improves the expression of existing inputs [19]. The CNN and DNN models utilized DeepChem to calculate MACCS key fingerprints and Morgan fingerprints, respectively, to train two predictive models. Then, we combined the two models to create a new HCD model. The HCD model combines the learning characteristics of CNN and DNN to enhance the data’s learning ability and improve prediction performance. After training and evaluation, we found that the HCD model had the best predictive performance compared to the CNN and DNN models. The accuracy (ACC), precision (PRE), recall, F-value, Matthews correlation coefficient (MCC), area under the receiver operating characteristic curve (AUROC), and area under the precision-recall curve (AUPRC) all significantly improved. The HCD model was able to better grasp the important characteristics of compounds and enhance predictive performance. After completing the model training, we validated its feasibility by testing it with 10 FDA-approved drugs known to promote platelet production. Of these drugs, seven were predicted to have drug activity, demonstrating the high reliability of the model’s predictive performance.

Our team previously developed a deep learning model using a combination of deep neural networks (DNNs) and recurrent neural networks (RNNs) to predict compounds that enhance megakaryocyte differentiation and platelet production. However, the model was trained on a limited dataset of only 350 samples. The number of samples used in the HCD model has more than doubled, which has enhanced the model’s generalization ability. Moreover, in terms of data processing, the MaxMin algorithm is applied to the dataset used in the HCD model to effectively reduce data similarity and significantly minimize the risk of model overfitting. When comparing the performance of hybrid DNNs with RNN and HCD, ACC (98.3%), PRE (97.0%), F-value (98.3%) and MCC (0.967) of HCD were higher than ACC (97.8%), PRE (95.7%), F-value (97.8%) and MCC (0.958) of hybrid DNN and RNN (HDR) models. Deep learning models trained on small datasets often suffer from overfitting, leading to limited generalization ability and mediocre predictive performance on unknown samples [46]. To assess the predictive capability of the HCD model in screening thrombopoietic drugs, we validated the model by obtaining 10 additional FDA-approved drugs for treating thrombocytopenia. We found that seven of them were predicted to be active. In addition, we conducted benchmark tests on the model to evaluate its performance and the reliability of its prediction results. It is evident that HCD outperforms other models in both aspects. In summary, the new hybrid deep learning model HCD improves data processing, refines the model architecture, optimizes model performance, and evaluates prediction accuracy.

Compounds with potential drug activity were virtually screened using an HCD model, and the activity of the compounds was subsequently verified through cell and animal experiments. One of these compounds, isochlorogenic acid A, promotes the differentiation and maturation of MK cells in K562 and Meg-01 cells. ICGA-A is a natural product derived from honeysuckle. The pharmacological study of ICGA-A is rare, but it is mainly related to antioxidant, antibacterial, anti-inflammatory, and hypoglycemic effects [47,48,49,50]. This study aimed to investigate the role of ICGA-A in platelet production.

Platelets are produced by MKs in the hematopoietic tissue of the BM, while MKs are produced by hematopoietic stem cells (HSCs) [51]. MKs undergo a complex process of differentiation and maturation to form platelets. This process includes multiple rounds of endomitosis, resulting in increased cytoplasmic content. Additionally, many depressions form on the surface of the cell membrane and protrude into the cytoplasm, forming an invaginated membrane system (IMS). The adjacent concave cell membranes fuse with each other in the deep part of the depression, causing a separation of part of the cytoplasm from the mother cell. Finally, these separated cytoplasmic fragments enter the blood circulation through the blood sinuses in the BM hematopoietic tissue, where they become platelets [52,53,54]. Newly produced platelets pass through the spleen, where they are partially stored, and are freely exchanged with platelets in the PB to maintain normal levels in the bloodstream [55,56]. Through experiments, we found that ICGA-A can increase the volume of K562 and Meg-01 cells, increase the number of nucleoli, and promote the expression of the surface-specific antigens CD41 (GPIIb/IIIa receptor) and CD42b (GPIb). These results indicate that ICGA-A can enhance the differentiation and maturation of MKs. To further investigate the therapeutic effect of ICGA-A on RIT, we conducted animal experiments and found that ICGA-A significantly enhanced platelet recovery in RIT mice. Furthermore, it had no impact on the levels of WBCs and RBCs. MKs are mainly derived from the BM [57]. H&E staining of and cytoploidy results for the BM showed that ICGA-A could increase the number of MKs in the BM and promote MK differentiation and maturation. The spleen is an important organ for extramedullary hematopoiesis (EMH) [58]. H&E staining of the spleen and cell ploidy revealed that, similarly to the BM, ICGA-A promoted the differentiation and maturation of splenic MKs. This finding provides further evidence for the role of ICGA-A in promoting an increase in platelet count. Most mature MKs are derived from the HSCs in the BM. HSCs produce MKs through a series of hematopoietic progenitor cell states [59]. Flow cytometry showed that ICGA-A increased the proportion of hematopoietic progenitors (CD41^−^CD117^+^), MK progenitors (CD41^+^CD117^+^), and MKs (CD41^+^CD117^−^). These results suggest that the increase in the number of MKs induced by ICGA-A is caused by promoting the formation and differentiation of hematopoietic progenitor cells into MKs. In RIT mice treated with ICGA-A, the expression of the MK markers CD41 and CD61 in the BM and spleen was significantly increased. This suggests that ICGA-A promotes the differentiation of MK in both the BM and spleen, which is consistent with the results of the in vitro experiments. In addition, we also detected the expression of CD41, CD61, and CD62P in PB. The results showed that after ICGA-A treatment, the number of PB platelets (CD41^+^CD61^+^) increased, while the production of abnormally activated platelets (CD41^+^CD62P^+^) was inhibited. In addition, we evaluated the toxicity of ICGA-A through H&E staining of the liver and kidney, as well as by measuring the body weight and organ index of mice. The results showed that ICGA-A had no toxic or side effects on RIT mice. Taken together, these data suggest ICGA-A is a potential drug for the treatment of RIT.

## 5. Conclusions

In summary, this study developed a virtual drug screening model using the DL algorithm to predict drugs with the potential to enhance MK differentiation and maturation, as well as platelet generation, from the compound library. The model automatically learns the characteristics of the compounds in the training set and predicts the activity of the drug based on those characteristics. Compared to the traditional drug screening model, the efficiency of new drug discovery is significantly improved, while the risk and cost are reduced. This study further confirms the reliability of the virtual screening model by predicting the activity of FDA drugs. Subsequently, the drug activity of the selected compounds was verified through in vitro and in vivo experiments. This study not only provides new ideas for model validation and new drug development, but also demonstrates that ICGA-A has the potential to be used as a drug for the clinical treatment of RIT.

## Figures and Tables

**Figure 1 biomolecules-14-00267-f001:**
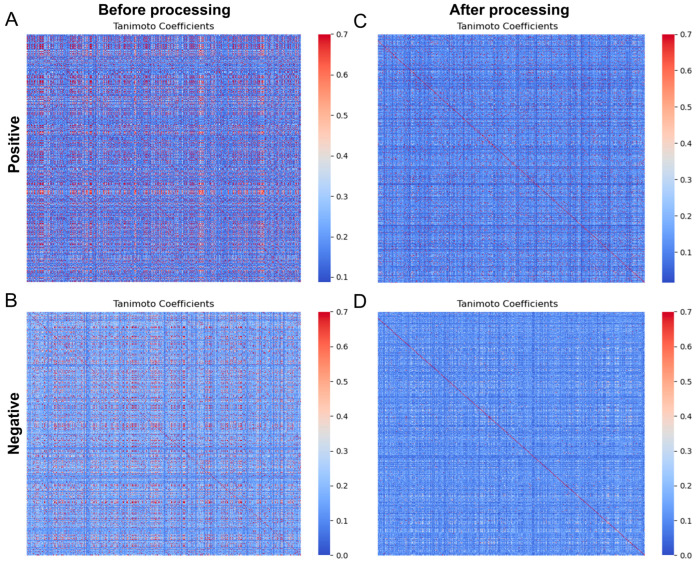
Tanimoto coefficient between compounds is reduced by the MaxMin algorithm. (**A**) Delete the positive sample before the highly similar sample. (**B**) Delete the positive sample after the highly similar sample. (**C**) Delete the negative sample before the highly similar sample. (**D**) Delete the negative sample before the highly similar sample.

**Figure 2 biomolecules-14-00267-f002:**
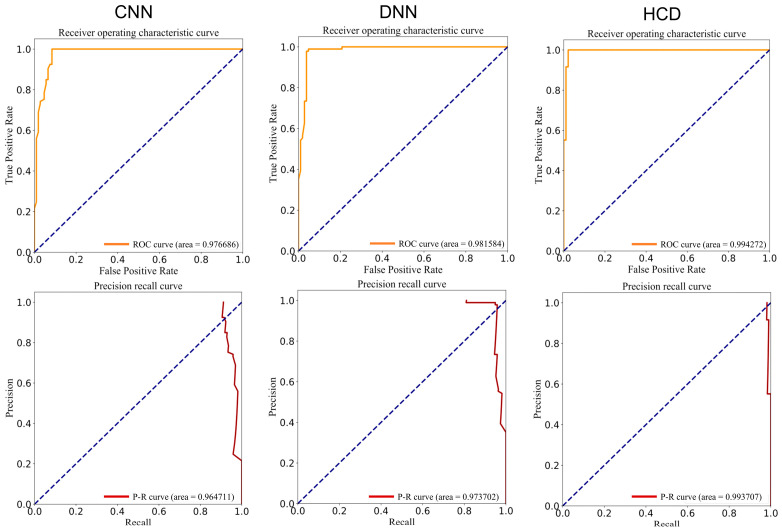
Receiver Operating Characteristic (ROC) and Precision-Recall (PR) curves generated by autoBioSeqpy for the three deep learning models.

**Figure 3 biomolecules-14-00267-f003:**
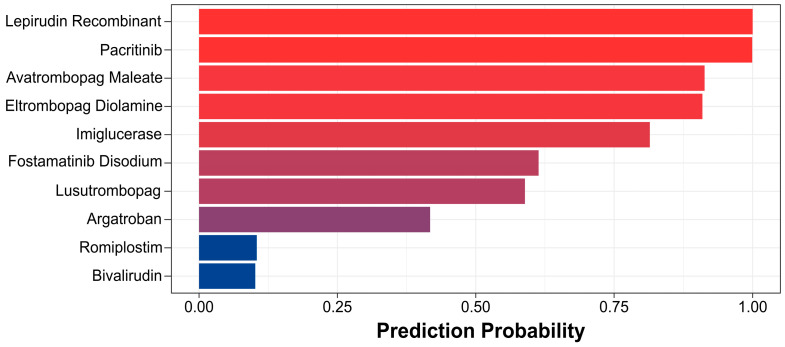
The HCD model validated by FDA drugs.

**Figure 4 biomolecules-14-00267-f004:**
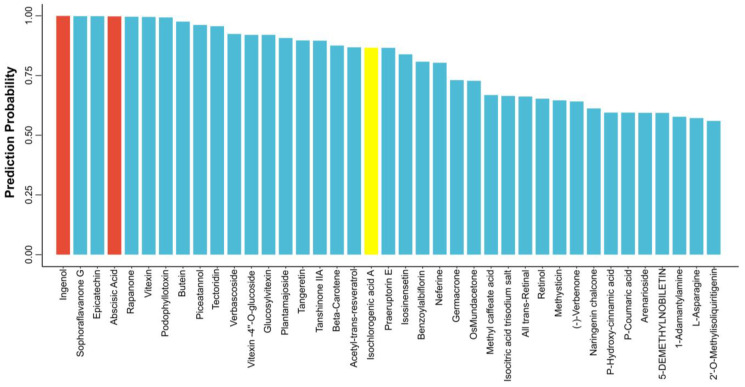
Compound molecules are ranked based on their prediction probability (39 compounds with scores > 0.5). Active compounds are marked in red, and potentially active compounds in yellow.

**Figure 5 biomolecules-14-00267-f005:**
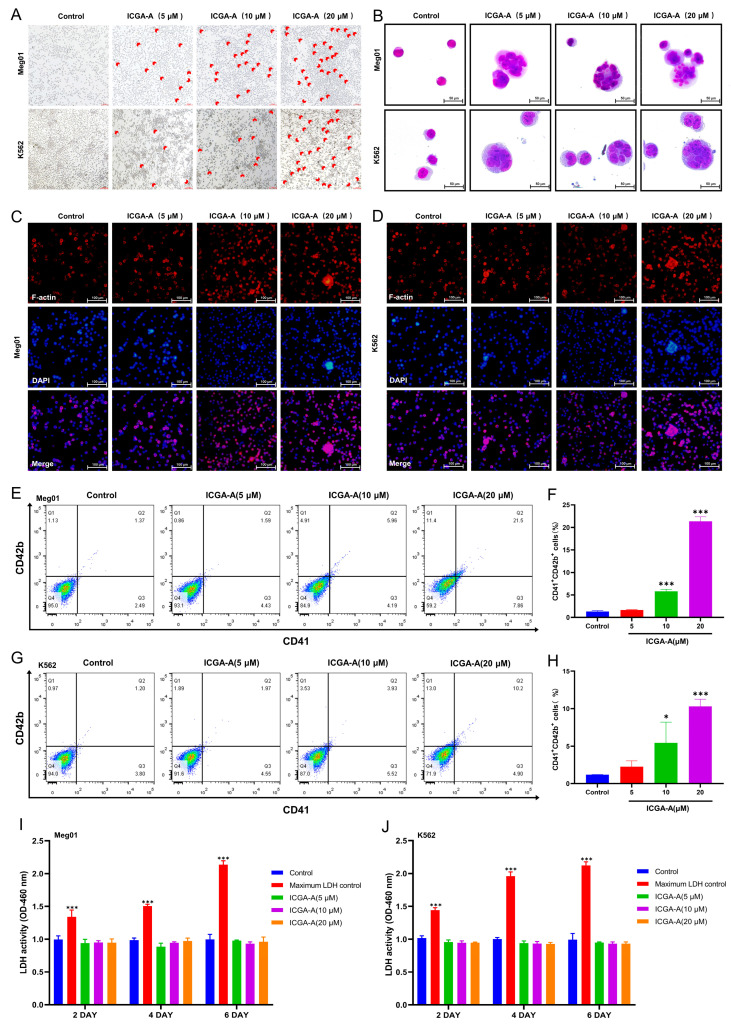
ICGA-A induces MK differentiation of Meg-01 and K562 cells. (**A**) Microscope photographs of Meg-01 and K562 cells treated with or without ICGA-A (5, 10 and 20 μM) for 5 days. Scale bar: 100 µm. (**B**) Giemsa staining of Meg-01 and K562 cells with or without ICGA-A (5, 10 and 20 μM) treatment for 5 days. Scale bar: 50 µm. (**C**,**D**) Phalloidin staining of Meg-01 and K562 cells with or without ICGA-A (5, 10 and 20 μM) treatment for 5 days. Scale bar: 100 μm. (**E**,**G**) CD41 and CD42b expression on Meg-01 and K562 cells with or without ICGA-A (5, 10 and 20 μM) treatment for 5 days. (**F**,**H**) The proportion of CD41+CD42b+ cells of Meg-01 and K562 cells in control- and ICGA-A-treated groups. Data are presented as mean ± SD (n = 3). (**I**,**J**) Detection of LDH activity of Meg-01 and K562 cells after treatment with or without ICGA-A (5, 10 and 20 μM) for 2, 4 and 6 days. Maximum LDH control represents the total amount of LDH present in the cells. Data are presented as mean ± SD (n = 3, ANOVA). * *p* < 0.05, *** *p* < 0.001 vs. the control group.

**Figure 6 biomolecules-14-00267-f006:**
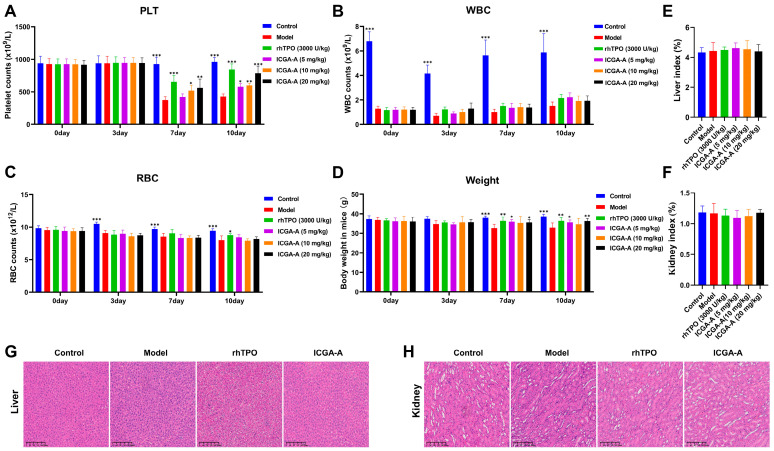
The therapeutic effects of ICGA-A on RIT mice. Blood routine examination in control group, model group, rhTPO group (3000 U/kg), low-dose ICGA-A group (5 mg/kg), medium-dose ICGA-A group (10 mg/kg), and high-dose (20 mg/kg) ICGA-A group on days 0, 3, 7, and 10. (**A**) Effect of ICGA-A on platelet levels. Data are mean ± SD (*n* = 6). (**B**) Effect of ICGA-A on WBC levels. Data are mean ± SD (*n* = 6). (**C**) Effect of ICGA-A on RBC levels. Data are mean ± SD (*n* = 6). (**D**) Effect of ICGA-A on body weight. Data are mean ± SD (*n* = 6). (**E**,**F**) Liver and kidney index of each group after 10 days of administration. Data are mean ± SD (*n* = 6, ANOVA). * *p* < 0.05, ** *p* < 0.01, *** *p* < 0.001 vs. the model group. (**G**,**H**) H&E staining of the liver and kidney in control, model, rhTPO (3000 U/kg) and ICGA-A treatment (20 mg/kg) groups on day 10. Bars: 200 µm.

**Figure 7 biomolecules-14-00267-f007:**
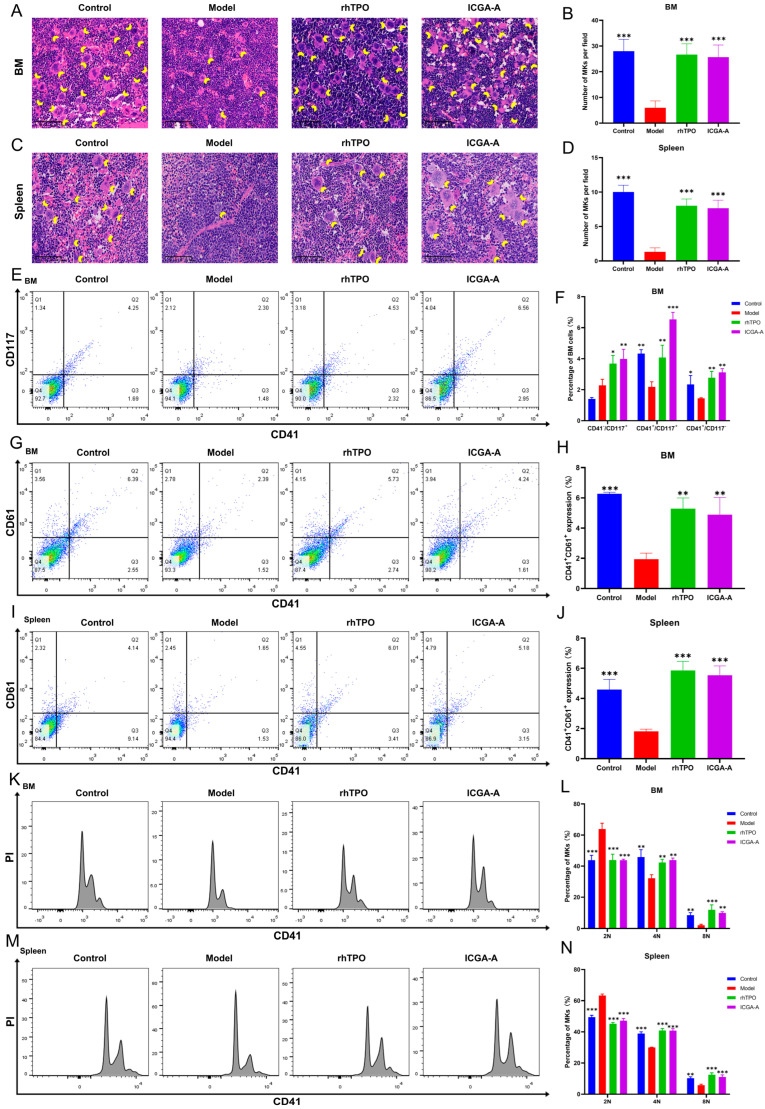
ICGA-A promotes MK differentiation and maturation in RIT mice. (**A**,**C**) H&E staining of BM and spleen in control, model, rhTPO (3000 U/kg) and ICGA-A treatment (20 mg/kg) groups on day 10. Bars: 100 µm. The MKs are indicated by arrows. (**B**,**D**) The MK counts of each group in BM and spleen. Data are mean ± SD (*n* = 3). (**E**) CD41 and CD117 expression in BM cells of each group on day 10. (**F**) The bar graph represents the percentage of CD41^−^CD117^+^, CD41^+^CD117^+^, and CD41^+^CD117^−^ cells of BM in each group. Data are mean ± SD (*n* = 3). (**G**,**I**) CD41 and CD61 expression in BM and spleen cells of each group on day 10. (**H**,**J**) The bar graph represents the percentage of CD41^+^ CD61^+^ cells of BM and spleen in each group. Data are mean ± SD (*n* = 3). (**K**,**M**) Ploidy analysis of the BM and spleen cells of each group on day 10. (**L**,**N**) The percentage of 2 N, 4 N and ≥ 8 N cells of BM and spleen in each group. Data are mean ± SD (*n* = 3, ANOVA). * *p* < 0.05, ** *p* < 0.01, *** *p* < 0.001 vs. the model group.

**Figure 8 biomolecules-14-00267-f008:**
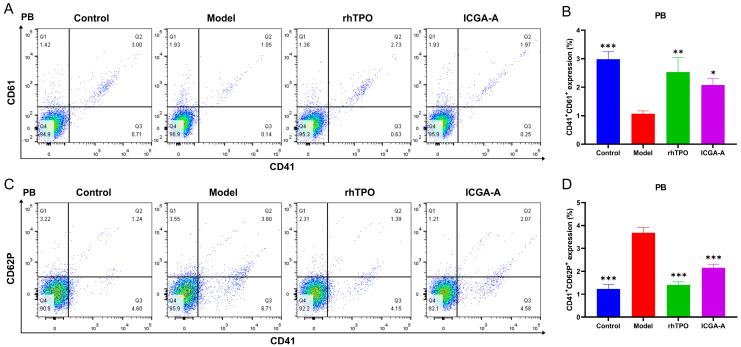
ICGA-A promotes platelet production and activation in RIT mice. (**A**) CD41 and CD61 expression in PB cells of each group on day 10. (**B**) The bar graph represents the percentage of CD41^+^ CD61^+^ cells of PB in each group. Data are mean ± SD (*n* = 3) (**C**) CD41 and CD62P expression in PB cells of each group on day 10. (**D**) The bar graph represents the percentage of CD41^+^CD62P^+^ cells of PB in each group. Data are mean ± SD (*n* = 3, ANOVA). * *p* < 0.05, ** *p* < 0.01, *** *p* < 0.001 vs. the model group.

**Table 1 biomolecules-14-00267-t001:** Performance of CNN, DNN, and HCD models.

Model	ACC (%)	PRE (%)	Recall (%)	F-Value (%)	MCC	AUROC	AUPRC
CNN	94.0	90.1	99.2	94.4	0.884	0.975	0.965
DNN	96.5	95.2	97.7	96.4	0.930	0.979	0.973
HCD	98.3	97.0	99.8	98.3	0.967	0.994	0.992

## Data Availability

The source codes, commands, and datasets used in this study are available at https://github.com/yta66/autoBioSeqpy/tree/master/examples/ICGA-A (accessed on 27 August 2023).

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
