# Peer review of "An Innovative Inducer of Platelet Production, Isochlorogenic Acid A, Is Uncovered through the Application of Deep Neural Networks"

_biomolecules, 2024, doi:10.3390/biom14030267_

Round 1
Reviewer 1 Report
Comments and Suggestions for Authors
This is a virtual screening approach actually followed by experimental validation, and therefore it certainly deserves to be published. My problem is that I cannot assess the practical importance of the discovery of "isochlorogenic acid" because I am no biologist/physician. I can only comment on the virtual screening part - which is successful... but overdone. Datasets of a few hundreds of compounds can be very well learned by out-of-box machine learners (random forest, support vector machines). At least, you should have started with that, sparing yourself the effort to build "deep" NNs which, as far as I can tell, learned to identify the structural patterns typical to redox-active compounds and radical traps. That is a rather easy task, any simple method would have done (as far as common sense can tell - if common sense is wrong please prove that in this particular case it's wrong!). Remember that in science SIMPLER models are always preferred at equal predictive power - unfortunately nobody follows this sane rule any more, and we are all fashion victims of deep learning, even if it is done to compute 2+2. If you want to "sell" the NN architectures as original, you'll have to do much more than this - a thorough benchmark against shallow machine learning approaches, over a large variety of property prediction challenges. Otherwise, all the NN-related details must be moved back to supplementary material, in order to keep the paper focused on the medicinal chemistry and biology - for that is where its potential strength lies (which, as I said, I cannot confirm as outsider to the field). It is an absolute no-go to print out obscure python commands - not even in supporting information. That is to be reported in user manuals: that would be the README file if you wish to publish your code on GitHub... after benchmarking it.
Comments on the Quality of English LanguageI am happy to finally see a vritual screening paper with experimental validation, BUT virtual screening (successful, indeed) is overdone (simpler out-of-box methods would have done as well, I guess) and I am not qualified to assess the biological relevance of the newly discovered compound - please send it to a reviewer in the field of cancer treatment.
Author Response
Dear Expert Reviewer,
Thank you very much for the prompt review process and excellent comments. We greatly appreciate the time and efforts which you have spent on it. We are submitting the revised manuscript entitled “An Innovative Inducer of Platelet Production, Isochlorogenic acid A, is Uncovered through the Application of Deep Neural Networks” (ID: biomolecules-2853843) to Biomolecules.
We have carefully considered your comments and suggestions and addressed each of the concerns in response to the comments (see point by point response). We have revised the manuscripts based on your comments and carefully checked throughout the manuscript and corrected the language errors. Our point-by-point responses to the comments (in blue) are shown below (in red and green).
"This is a virtual screening approach actually followed by experimental validation, and therefore it certainly deserves to be published. My problem is that I cannot assess the practical importance of the discovery of "isochlorogenic acid" because I am no biologist/physician. I can only comment on the virtual screening part - which is successful... but overdone. Datasets of a few hundreds of compounds can be very well learned by out-of-box machine learners (random forest, support vector machines). At least, you should have started with that, sparing yourself the effort to build "deep" NNs which, as far as I can tell, learned to identify the structural patterns typical to redox-active compounds and radical traps. That is a rather easy task, any simple method would have done (as far as common sense can tell - if common sense is wrong please prove that in this particular case it’s wrong!). Remember that in science SIMPLER models are always preferred at equal predictive power - unfortunately nobody follows this sane rule anymore, and we are all fashion victims of deep learning, even if it is done to compute 2+2. If you want to "sell" the NN architectures as original, you'll have to do much more than this - a thorough benchmark against shallow machine learning approaches, over a large variety of property prediction challenges. Otherwise, all the NN-related details must be moved back to supplementary material, in order to keep the paper focused on the medicinal chemistry and biology - for that is where its potential strength lies (which, as I said, I cannot confirm as outsider to the field). It is an absolute no-go to print out obscure python commands - not even in supporting information. That is to be reported in user manuals: that would be the README file if you wish to publish your code on GitHub... after benchmarking it."
Response: Thank you very much for your excellent comments and suggestions. I agree with you that, all else being equal, simple models are preferred. After completing the training and testing of the machine learning model, we assessed its generalization ability using 10 FDA-approved drugs for thrombocytopenia. The results showed that the top-performing neural network model could only accurately predict the activity of two drugs. In the article, we mentioned that the deep learning model HCD can predict 7 drug activities, and its generalization ability is much stronger than that of the machine learning model, and evaluation metrics of HCD are also superior to those of other models (Table 2 and 3). The use of AI in drug prediction aims to accurately and efficiently discover new drugs, reduce the cost of drug research and development, and shorten the research and development cycle for new drugs [1]. Therefore, we chose to use the HCD model for its superior generalization and performance. I agree with you that typing python commands can be a significant challenge for non-experts. But autoBioSeqpy is a tool that uses deep learning for biological sequence classification. The advantage of this tool is its simplicity. Users only need to prepare the input data set and then use a command line interface. Then, autoBioSeqpy automatically executes a series of customizable steps including text reading, parameter initialization, sequence encoding and model loading, training, and testing. In short, the python commands provided in this article serve as templates for non-experts. All they need to do is ensure that their dataset name matches the dataset name in the Python command, and then copy the python command into autoBioSeqpy. We have published python commands, model programs, datasets, and usage methods on GitHub. Link to https://github.com/yta66/autoBioSeqpy/tree/master/examples/ICGA-A. Please refer to the highlighted section on page 3, 9 and 10.
(The highlighted section on page 3):
In addition, we trained five machine learning models using software orange3(https://orangedatamining.com/download/): Support Vector Machine (SVM), Random Forest (RF), Neural Network (NN), Logistic Regression (LR), and Adaptive Boosting (AB). We used MACCS keys fingerprint, Morgan fingerprint, and a combination of MACCS keys fingerprint with Morgan fingerprint as their inputs.
(The highlighted section on page 9):
In addition, the ACC, PRE, recall, F-value, and AUROC of the deep learning model HCD are all higher than those of the five machine learning models (Supplement Table 1).
(The highlighted section on page 10):
while the machine learning model could predict that up to two drugs would be active (Supplement Table 2).
Supplement Table 1 & 2 please find in the attachment file.
Reference:
- Blanco-González, A., A. Cabezón, A. Seco-González, D. Conde-Torres, P. Antelo-Riveiro, Á. Piñeiro and R. Garcia-Fandino. "The Role of AI in Drug Discovery: Challenges, Opportunities, and Strategies." Pharmaceuticals (Basel) (2023) 16(6).https://doi.org/10.3390/ph16060891
Thank you for all the valuable and helpful comments and suggestions.
Best regards,
Jianming Wu

Reviewer 2 Report
Comments and Suggestions for Authors
In this manuscript, Yi et al. present a virtual screening approach based on a deep learning classification tool that leads to identifying a set of putative molecules with activity in inducing platelet production. Cellular tests show that one such predicted compound, previously unknown, is active on two different cell lines. In vivo essays in mice demonstrate the activity in recovering platelet formation.
I believe this is an interesting study that deserves publication, provided that the following questions are addressed:
1) The comparison between different neural networks is presented with some details as far as results and performance, but not enough detail is given regarding architecture. Could the authors add a workflow diagram of the different NNs to highlight their architecture (e.g. number and kind of layers, etc) if they want to comment on the performance? I understand that the used tool offers predefined architectures and was only applied and not developed in this manuscript, but, given the “tutorial” flavor of their Methods section I believe it would add nicely to the performance discussion.
2) A structural analysis of the training set and of the predicted compounds, before the experimental validation, should be added to try and understand the variety of chemical species that the model predicts, to evaluate the innovation of the approach.
3) There is a very similar paper by the same authors, recently published and cited in this manuscript, namely: Mo, Q., T. Zhang, J. Wu, L. Wang and J. Luo. "Identification of thrombopoiesis inducer based on a hybrid deep neural network 692 model." Thromb Res (2023) 226: 36-50. The authors should stress in the discussion all the relevant differences (dataset construction, performance of the published NN on the same datasets) that justify the development of the current approach, and discuss the results in this respect.
Comments on the Quality of English LanguageEnglish language is fine
Author Response
Dear Expert Reviewer,
Thank you very much for the prompt review process and excellent comments. We greatly appreciate the time and efforts which you have spent on it. We are submitting the revised manuscript entitled “An Innovative Inducer of Platelet Production, Isochlorogenic acid A, is Uncovered through the Application of Deep Neural Networks” (ID: biomolecules-2853843) to Biomolecules.
We have carefully considered your comments and suggestions and addressed each of the concerns in response to the comments (see point by point response). We have revised the manuscripts based on your comments and carefully checked throughout the manuscript and corrected the language errors. Our point-by-point responses to the comments (in blue) are shown below (in red and green).
1."The comparison between different neural networks is presented with some details as far as results and performance, but not enough detail is given regarding architecture. Could the authors add a workflow diagram of the different NNs to highlight their architecture (e.g. number and kind of layers, etc) if they want to comment on the performance? I understand that the used tool offers predefined architectures and was only applied and not developed in this manuscript, but, given the “tutorial” flavor of their Methods section I believe it would add nicely to the performance discussion".
Response: Thank you for your rigorous thinking and constructive suggestion. Based on your suggestion, the workflow diagrams for the three neural network models utilized in this study were created to illustrate their architectures, and the details of each layer of the model were thoroughly explained (Supplement Figure 1). Please refer to the highlighted section on page 3.
(The highlighted section on page 3):
Detailed workflow diagrams for the CNN, DNN, and HCD models are shown in Supplementary Figure 1.
Supplement Figure 1 please find in the attachment file.
Supplement Figure 1. Workflow diagrams of CNN, DNN, and HCD models. (a) The input to the DNN model is a 2048-bit Morgan fingerprint. (b) The first dense layer of the DNN model consists of 64 neurons activated by a ReLU activation function, with a dropout rate of 0.2. (c) The second dense layer of the DNN model consists of 64 neurons activated by a ReLU activation function, with a dropout rate of 0.2. (d) The output layer of the DNN model consists of a neuron activated by a sigmoid function. (e) Input 167-bit MACCS key fingerprints into the embedding layer of the CNN model to convert positive integers into dense vectors of a fixed size, with a dropout rate of 0.2. (f) The convolutional layer of the CNN model is activated by the ReLU activation function, with a convolutional window length of 16, a stride length of 1, and it utilizes 330 filters. (g) The pooling layer of the CNN model inputs the maximum feature value extracted from the filter into the next layer. (h) The dense layer of the CNN model consists of 128 neurons activated by a ReLU activation function, with a dropout rate of 0.2. (i) The output layer of the CNN model consists of a neuron activated by a sigmoid function. (j) Remove the output layer from the CNN and DNN models. The outputs of the remaining layers are connected and fed into an output layer consisting of a single neuron, which is activated by the sigmoid function.
2. "A structural analysis of the training set and of the predicted compounds, before the experimental validation, should be added to try and understand the variety of chemical species that the model predicts, to evaluate the innovation of the approach."
Response: Thanks so much for your constructive suggestion and kindly guidance. According to your suggestion, before the experimental verification, we calculated the eigencoefficients and conducted UMAP analysis between the training set and the predictive compound ICGA-A to assess the novelty of the method and ICGA-A. Please refer to the highlighted section on page 10-11.
(The highlighted section on page 10-11):
In addition, we calculated a Tanimoto Coefficient between the compound in the training set and ICGA-A (Supplement Figure 2). We did not find highly similar compounds in the training set. A UMAP analysis was then conducted, revealing that ICGA-A was similar to the training set in chemical space defined by the MACCS keys and Morgan fingerprint (Supplement Figure 3). This finding demonstrates that deep learning models can uncover the underlying structure of compounds. The above results show that ICGA-A is a novel compound with the potential to promote megakaryocyte differentiation and platelet formation.
Supplement Figure 2 please find in the attachment file.
Supplementary Figure 2. Compounds with the top 50 Tanimoto Coefficient between the training set compounds and ICGA-A
Supplement Figure 3 please find in the attachment file.
Supplementary Figure 3. The chemical space defined by the MACCS keys and Morgan fingerprint with UMAP analysis.
3. "There is a very similar paper by the same authors, recently published and cited in this manuscript, namely: Mo, Q., T. Zhang, J. Wu, L. Wang and J. Luo. "Identification of thrombopoiesis inducer based on a hybrid deep neural network 692 model." Thromb Res (2023) 226: 36-50. The authors should stress in the discussion all the relevant differences (dataset construction, performance of the published NN on the same datasets) that justify the development of the current approach and discuss the results in this respect."
Response: Thanks a lot for the excellent comments and suggestions. We will incorporate your suggestions in the discussion section of this article to explore the variances from previous deep learning models and highlight the enhancements and prospects of current models. Please refer to the highlighted section on page 17-18 and reference 46 for the details.
(The highlighted section on page 17-18 and reference 46):
Our team previously developed a deep learning model using a combination of deep neural networks (DNNs) and recurrent neural networks (RNNs) to predict compounds that enhance megakaryocyte differentiation and platelet production. However, the model was trained on a limited dataset of only 350 samples. The number of samples used in the HCD model has more than doubled, which has enhanced the model's generalization ability. Moreover, in terms of data processing, the MaxMin algorithm is applied to the dataset used in the HCD model to effectively reduce data similarity and significantly minimize the risk of model overfitting. When comparing the performance of hybrid DNNs with RNN and HCD, ACC (98.3%), PRE (97.0%), F-value (98.3%) and MCC (0.967) of HCD were higher than ACC (97.8%), PRE (95.7%), F-value (97.8%) and MCC (0.958) of hybrid DNN and RNN (HDR) models. Deep learning models trained on small datasets often suffer from overfitting, leading to limited generalization ability and mediocre predictive performance on unknown samples[1]. To assess the predictive capability of the HCD model in screening thrombopoietic drugs, we validated the model by obtaining 10 additional FDA-approved drugs for treating thrombocytopenia. We found that 7 of them were predicted to be active. In addition, we conducted benchmark tests on the model to evaluate its performance and the reliability of its prediction results. It is evident that HCD outperforms other models in both aspects. In summary, the new hybrid deep learning model HCD improves data processing, refines the model architecture, optimizes model performance, and evaluates prediction accuracy.
Reference:
- Alzubaidi, L., J. Bai, A. Al-Sabaawi, J. Santamaría, A. S. Albahri, B. S. N. Al-dabbagh, M. A. Fadhel, M. Manoufali, J. Zhang, A. H. Al-Timemy, et al. "A survey on deep learning tools dealing with data scarcity: definitions, challenges, solutions, tips, and applications." Journal of Big Data (2023) 10(1): 46.https://doi.org/10.1186/s40537-023-00727-2
Thank you for all the valuable and helpful comments and suggestions.
Best regards,
Jianming Wu

Round 2
Reviewer 2 Report
Comments and Suggestions for Authors
The authors have addressed all my requests